# Scaling Laws for Sparsely-Connected Foundation Models

**Elias Frantar**[1,2]**, Carlos Riquelme**[1]**, Neil Houlsby**[1]**, Dan Alistarh**[2]**, Utku Evci**[1]

[1] Google DeepMind, [2] ISTAustria; correspondence: `elias.frantar@ist.ac.at`

## Abstract

We explore the impact of parameter sparsity on the scaling behavior of Transformers trained on massive datasets (i.e., "foundation models"), in both vision and language domains. In this setting, we identify the first scaling law describing the relationship between weight sparsity, number of non-zero parameters, and amount of training data, which we validate empirically across model and data scales; on ViT/JFT-4B and T5/C4. These results allow us to characterize the "optimal sparsity", the sparsity level which yields the best performance for a given effective model size and training budget. For a fixed number of non-zero parameters, we identify that the optimal sparsity increases with the amount of data used for training. We also extend our study to different sparsity structures (such as the hardware-friendly n:m pattern) and strategies (such as starting from a pretrained dense model). Our findings shed light on the power and limitations of weight sparsity across various parameter and computational settings, offering both theoretical understanding and practical implications for leveraging sparsity towards computational efficiency improvements. We provide pruning and scaling law fitting code at: `github.com/google-research/jaxpruner/tree/main/jaxpruner/projects/bigsparse`.

## 1 Introduction

Foundation models (Bommasani et al., 2021), loosely defined as large (often Transformer-based (Vaswani et al., 2017)) networks that are trained on massive quantities of highly general data, have driven significant progress in deep learning, for both natural language (Brown et al., 2020) and vision tasks (Dosovitskiy et al., 2021). One key property of such models is the predictability of their performance when scaling various model attributes, such as the number of parameters and the amount of data or computation used (Kaplan et al., 2020). This is encapsulated by *scaling laws*, which make it possible to accurately predict the final performance of a model given its size and training budget (i.e., amount of training data).

A parallel trend, motivated by computational costs, has been the focus towards increased efficiency for large models. This is usually achieved by employing compressed parameterizations via quantization (Gholami et al., 2021) or sparsification (Hoefler et al., 2021), during inference and/or training, which can lead to reduced run-time via both software and hardware support (Elsen et al., 2020; Yao et al., 2022). Despite major community interest in efficiency, the impact of these compressed representations, in particular of parameter/weight sparsity, on the scaling behavior of foundation models is not well understood; especially, when applying powerful but expensive training-based compression methods (Jacob et al., 2018; Zhu & Gupta, 2017).

In this paper, we address this gap by studying the relationship between sparsity and scaling laws for foundation models. We focus on *weight sparsity*, that is, on networks whose individual connections are pruned, and on Transformer (Vaswani et al., 2017) models for both vision (Dosovitskiy et al., 2021) and language (Raffel et al., 2020b) domains. We use the massive JFT-4B (Google, 2023a) and C4 (Raffel et al., 2020a) datasets, which are several orders of magnitude larger than what has been employed so far by the vast majority of work on sparsity. In this massive dataset regime, dense models continue to improve with prolonged training and it is unclear if sparse models can be competitive in a fair comparison, using equal amounts of training compute and data. This is in contrast to popular pruning benchmarks, e.g., ImageNet (Deng et al., 2009), executing for many training epochs, where dense models tend to saturate (Kuznedelev et al., 2023b), allowing sparse models to achieve major gains relative to dense models with a comparable number of parameters.

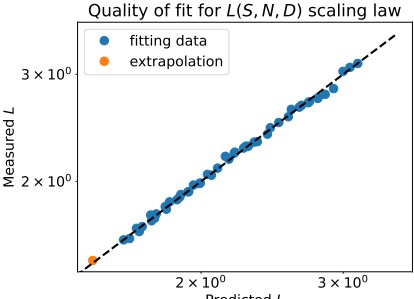
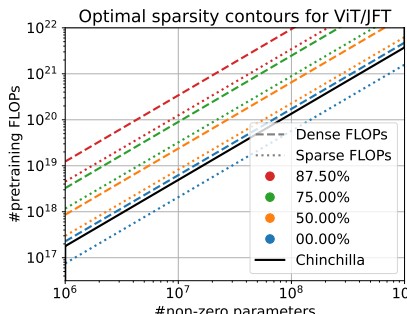

Figure 1: (Left) Fit and extrapolation quality of the $L(S, N, D)$ scaling law on T5/C4. (Right) Optimal sparsity $S_{\text{opt}}$ contours fitted on ViT/JFT, for sparse and dense costs (details in Section 2.2).

In order to quantify the benefits of sparsity, or the lack thereof, in this large-dataset regime we develop joint scaling laws that relate the sparsity of a network, its effective size and the amount of data used for training. We show that, for sparsity $S$, number of non-zero parameters $N$ and amount of training data/steps $D$, the validation loss $L$ approximately satisfies the following law, for both vision and language tasks:

$$L(S, N, D) = \left(a_S(1 - S)^{b_S} + c_S\right) \cdot \left(\frac{1}{N}\right)^{b_N} + \left(\frac{a_D}{D}\right)^{b_D} + c, \tag{1}$$

Intuitively, the first two summands capture the power law scaling in terms of capacity, i.e. sparsity and non-zero parameters, and data, respectively, while $c$ is a lower bound on the achievable task loss. In more detail, the first multiplicative term captures the impact of sparsity, here expressed as remaining density $(1 - S)$, which itself follows a saturating power-law with coefficient $a_S$, exponent $b_S$ and limit constant $c_S$. The exponents $b_N$ and $b_D$ scale the (non-zero) parameter count $N$, and the data $D$ term, respectively, as is common in classical scaling laws (Kaplan et al., 2020).

We validate this formula empirically using large vision and language datasets, several model sizes, amounts of training data and sparsity levels. Please see Figure 1 (Left) for an illustration of the scaling law fit and extrapolation quality. In turn, this law allows us to obtain several new insights for sparsely connected foundation models:

- First, the sparse scaling law suggests that sparsity affects each model size in a similar way, i.e., as a multiplicative constant to the size scaling. At the same time, sparsification does not appear to interact significantly with the data scaling; the original dense term in $D$ is preserved.

- Second, we can use our scaling law in Equation (1) to analytically derive the *optimal sparsity* $S_{\text{opt}}$ for a given inference size and training budget, allowing us to predict the regime where sparsity could actually provide benefits over simple dense model rescaling and extended training.

- Our analysis of optimal sparsity $S_{\text{opt}}$, demonstrated in Figure 1 (Right), shows that its iso-contours run parallel to the dense compute optimal Chinchilla line (Hoffmann et al., 2022) of the respective model and task. Importantly, the optimal sparsity increases with longer training. Further, while optimal dense models define a line on the parameter-FLOPs surface, optimal sparse models form a half-plane (with different sparsities unlocking multiple optimal sizes for a fixed training cost).

- In addition, we find that the main conclusions of our law hold also for hardware-friendly n:m sparsity patterns (Mishra et al., 2021), that relative capacity gains through sparsity are consistent across domains, and that pruning well-trained dense models is more efficient than training from scratch (while sparsifying), if dense checkpoints already exist, but is significantly slower otherwise.

In sum, our results provide the first scaling law for characterizing the impact of sparsity on the performance of Transformers trained on massive datasets. From the conceptual perspective, this provides a simple tool to understand the power–but also the limitations–of sparsity for a given task/model combination. From the practical side, this can be used to determine whether sparsity can be a reasonable option for inference or training speedups, in settings where specific software/hardware support for such compressed representations is available.

## 2 SCALING LAWS FOR PARAMETER-SPARSE TRANSFORMERS

**Fair evaluation in the presence of strong scaling.** In the context of modern Transformers trained on massive datasets, popular evaluation approaches for pruning (Gale et al., 2019; Singh & Alistarh, 2020; Sanh et al., 2020; Schwarz et al., 2021; Benbaki et al., 2023) that have been reasonable for standard benchmarks like ResNet50/ImageNet (Singh & Alistarh, 2020; Schwarz et al., 2021) or BERT/GLUE (Sanh et al., 2020; Kurtic et al., 2022), require careful reconsideration to ensure meaningful comparisons. Specifically, it is critical to adopt the resource-equivalent setting of Liu et al. (2018) and Jin et al. (2022), which we illustrate below.

For example, assume that a model pretrained for 100k steps is pruned to 50% sparsity over another 100k steps (a standard setup for ResNet50/ImageNet). The resulting network should not be compared to the original one, as it has had $2\times$ more training overall. Further, even a comparison against a $2\times$ smaller model (same non-zero parameter count) trained for 200k steps (same amount of training) is not necessarily fair, as training this smaller dense model requires less overall compute than producing the larger sparse one (as we perform sparsification only gradually). In both cases, due to the strong scaling properties of Transformers trained on massive quantities of data (Kaplan et al., 2020; Hoffmann et al., 2022), the respective baseline would have most likely improved significantly with more data/compute as well. Thus, the proper comparison point for a sparse network is a *dense model with the same number of parameters trained for equivalent compute*. This resource normalization, required by strong scaling and no overfitting, renders this setting very challenging.

**Experimental setup overview.** We execute extensive training sweeps across sparsity, size and data, which we then subsequently use to develop scaling laws. Now follows a very brief summary of our main setup; a detailed discussion of all our choices, including the experiment grid and hyperparameters, can be found in Appendix A. In terms of models and datasets, we focus on Vision Transformers (Dosovitskiy et al., 2021) trained for multi-label image classification on the JFT-4B dataset (Dehghani et al., 2023), consisting of 4 billion images, as well as encoder-decoder T5 models (Raffel et al., 2020b) (improved 1.1 version (Google, 2023b)) trained for masked-language-modelling on C4 (Raffel et al., 2020b), consisting of 150+ billion tokens. We follow the model's respective original training recipes (Zhai et al., 2022; Raffel et al., 2020b) and carry out sparsification *during* training via gradual magnitude pruning (Zhu & Gupta, 2017), using a cubic schedule starting at 25% of training and ending at 75%. Our setup is optimized for robustness and consistency across scales rather than maximizing pruning performance on one particular setting (see also Appendix B).

### 2.1 DERIVING THE CORE LAW

**Dense scaling.** It is well established (Kaplan et al., 2020; Hoffmann et al., 2022) that the pretraining validation loss of *dense* Transformers can be approximately modeled, in terms of parameter count $N$ and amount of training data $D$, by functions of the following form:

$$L(N, D) = \left(\frac{a_N}{N}\right)^{b_N} + \left(\frac{a_D}{D}\right)^{b_D} + c. \tag{2}$$

The first two summands capture the power law scaling in terms of size and data, respectively. Meanwhile, $c$ represents the inherent stochasticity of the modelling problem as a lower bound on the loss. The scaling exponents $b_N$ and $b_D$ are usually quite stable for a particular task, whereas the constant coefficients $a_N$ and $a_D$ vary with minor process changes like a different architecture or optimizer.

Scaling laws usually assume an ideal training setup with no data repetition and focus on modelling the non-bottlenecked regime (e.g., with sufficient steps/data/batchsize/etc.) rather than on edge cases (Kaplan et al., 2020; Hoffmann et al., 2022); we follow suit. Further, we deliberately consider the pretraining loss and infinite data setting to assess the effectiveness of sparsity in its most challenging (one essentially needs to fit the data as well as possible) yet also most useful application (all further post-processing would directly benefit from a compressed base model; see also Appendix E).

**Preliminary observations.** The key question we hope to address is how parameter sparsity $S$ enters this core scaling relationship; understanding this will enable studying other interesting aspects like optimal sparsity or limit performance. A priori, it is not obvious how $S$ should enter into Equation (2) to form $L(S, N, D)$, where $N$ denotes the number of *non-zero parameters*. Are larger models easier to sparsify, does longer training help highly sparse models more, or is sparsity mostly

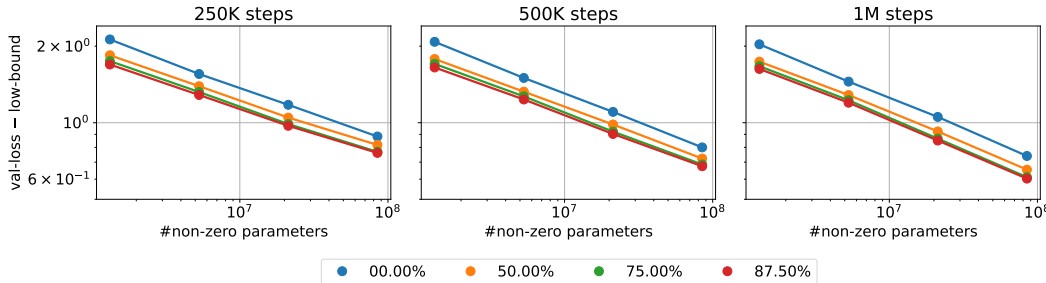

Figure 2: Visualization of T5/C4 sweep results for all sizes and sparsities, grouped by training steps.

independent of other parameters? Therefore, to identify the functional form of the scaling law, we run a T5 training sweep over sparsity, size and steps. Figure 2 shows validation loss (with a lower bound $c = 1$ subtracted to account for power law saturation against the inherent uncertainty limit) versus model size for all sparsity levels, grouped by the number of training steps. Please observe that the scaling of this plot, as well as most other visualizations in this paper, is log-log.

We make three major observations from these graphs:

1. The loss vs. #non-zero curves for all sparsity levels seem to form almost parallel lines, differing primarily in the intercept.
2. The higher the sparsity the lower the loss, but gains are quickly diminishing.
3. The overall shape of all curves is very similar for each training duration, the y-values just tend to shift a bit downwards with more training steps.

**Sparse scaling law.** We now use the previous insights to construct our $L(S, N, D)$ formula. Observation 1 suggests that the model size power law scaling for all sparsity levels differs primarily by a constant factor (intercept in a log-log plot); $b_N$ (the slope) stays fairly consistent. Based on observation 2, we model this sparsity factor as a (quickly) saturating power law. Finally, observation 3 indicates that sparsity and data scaling are mostly independent, hence we simply keep the original $D$-term. In summary, these observations lead us to the following joint scaling law:

$$L(S, N, D) = \left(a_S(1-S)^{b_S} + c_S\right) \cdot \left(\frac{1}{N}\right)^{b_N} + \left(\frac{a_D}{D}\right)^{b_D} + c. \tag{3}$$

To properly model that $0.75$ is twice as sparse as $0.5$, we define the sparsity power-law part via the corresponding compression rate $1/(1 - S)$. Further, $a_N$ is subsumed by $a_S$ and $c_S$, leaving 7 free parameters. On a high level, our scaling law combines a *capacity limit* term, comprised of size and sparsity (which can encode extra information via its zero pattern), with the standard data limit term. Lastly, $S = 0$ recovers the established $L(N, D)$ form.

**T5/C4 results.** Next, we fit the coefficients of $L(S, N, D)$ to our entire T5 sweep data. This is accomplished, following (Hoffmann et al., 2022), by minimizing the Huber-loss of $\log L$ with $\delta = 0.001$ (for robustness against outliers) using BFGS, for multiple random starting points. We plot actual vs. predictions in Figure 1 (Right) to judge the quality of our final fit (see Appendix C for coefficient values). All in all, the predictions match the observed data quite closely (despite having $\approx 7$ datapoints per free parameter), demonstrating the compatibility of the law in (3) with the observations.

Furthermore, we evaluate extrapolation performance by pruning a 2.3 billion parameter model to $75\%$ sparsity. This constitutes an $\approx 6.75\times$ *larger* target number of non-zero parameters than the maximum in our fitting data, which is a similar level of extrapolation as was done in the Chinchilla study (Hoffmann et al., 2022). To avoid any architecture bottlenecks and achieve better training utilization, we use the T5-XL architecture (rather than a rescaled T5-base) and train with batchsize 256 for 250k steps (rather than 500k with batchsize 128). Despite these changes to our setup, the prediction of our fitted scaling law is quite close to the actual validation loss; see Figure 1 (Right).

**ViT/JFT-4B results.** Lastly, we execute a ViT training sweep and also fit a scaling law of the same (3) form as for the T5 data. Here we use $\delta = 0.01$ and do not take the log of $L$ as we find the

NLP-optimized settings from before to exclude outliers too aggressively for ViT data (which gives a poor fit for smaller models). We note that this sweep contains $> 2\times$ more datapoints, leading to more robust coefficient estimates. We qualitatively compare predicted and actual loss-vs-data curves in Figure 3, organized by sparsity level. We strongly emphasize that the predictions in all subplots here are produced by *a single joint law* with the same parameters (*not* one fit per image). As can be seen, for the most part, our law appears to match the collected datapoints very well. Only at the lowest amount of training, some points are a bit off the prediction curve; we suspect that this may be related to the fact that these runs only involve comparatively few training steps, which may be a slight bottleneck for the optimization process. Finally, we train a 75% ViT-L for 3.6 billion images to validate extrapolation, ending up only $0.09$ better than the corresponding prediction, in line with being a slightly better base architecture than the family used for our sweep (see Appendix A).

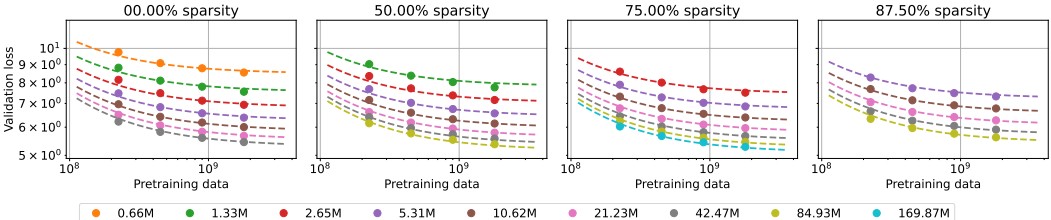

Figure 3: Visual comparison of the ViT scaling sweep data and the corresponding fitted scaling law.

## 2.2 OPTIMAL SPARSITY

One particularly interesting feature of the joint scaling law just derived is that it allows easily comparing models with different sparsities but the same number of non-zero parameters and training cost. Thus, we can determine in which situations sparse models are better than dense ones, according to all criteria discussed in Section 2. Specifically, we can define the following quantity:

**Optimal sparsity.** *The sparsity value $S_{opt}(N, C)$ which yields the lowest validation loss for a fixed number of non-zero parameters $N$ and training cost $C$.*[1]

There are two ways of defining training costs in this context: (a) *densely*, as the cost of training a dense base model of size $N/(1-S)$ for the same amount of training steps, or (b) *sparsely*, as the actual FLOPs spent to produce the sparse model, assuming that sparsity can be perfectly exploited during training as soon as it appears. For our particular sparsification schedule, (b) can be calculated by multiplying the training costs of a dense model, approximated as $6ND$ (Kaplan et al., 2020) (or half for encoder-decoder architecture models), by (see Appendix D for this and other derivations):

$$c_{\text{mul}}(S) = (0.25 + 0.50 \cdot (1 - 0.75 \cdot S))/(1 - S) + 0.25. \tag{4}$$

As we have assumed that the amount of training equals the amount of new data, we can determine the performance of a sparsity $S$ model trained for compute $C = 6ND \cdot c_{\text{mul}}(S)$ by querying $L$ with $D_S = (C/6N)/c_{\text{mul}}(S)$, i.e., scaling down the $D$ corresponding to $C$ by the increase in training costs of the sparse model. Inserting $D_S$ and then differentiating with respect to $S$ gives the contour line for which sparsity $S$ is optimal, i.e., achieves the lowest loss among all possible sparsity choices, when training for the same compute:

$$b_D \cdot \frac{c'_{\text{mul}}(S)}{c_{\text{mul}}(S)} \cdot \left(c_{\text{mul}}(S) \cdot \frac{a_D}{D_S}\right)^{b_D} = a_s b_s (1 - S)^{b_S - 1} \cdot N^{-b_N}. \tag{5}$$

An interesting property about this contour is that it implies $D_S = O(N^{b_N/b_D})$, meaning that if data- is stronger than size-scaling, then the same sparsity is optimal for a smaller data-to-size ratio on larger models. This is sensible as a process bottlenecked more by capacity than by data will benefit more from increasing the former, e.g., by adding sparsity. Finally, we want to point out that $S_{\text{opt}}$ can often also be determined explicitly by solving (4) for $S$, e.g., here for dense training costs with $c_{\text{mul}}(S) = 1/(1 - S)$:

$$S_{\text{opt}}(N, C) = \max\left\{1 - \exp\left[\left(\log\frac{b_D}{a_S b_S} + b_N \log N\right)/(b_D + b_S)\right] \cdot \left(\frac{6 a_D N}{C}\right)^{b_D/(b_D + b_S)}, 0\right\}. \tag{6}$$

---

[1]We note that it is common in the literature (Hoffmann et al., 2022) to define scaling laws in terms of parameters $N$ and data $D$, but switch to expressing scaling in terms of computational cost $C$ whenever relevant.

**Empirical results.** We now compute optimal sparsity curves for our experimental T5 and ViT data, for which we fit scaling laws in the previous subsection. Figure 1 (Right) and 4 show the optimal sparsity contours, both for dense and sparse costs. An interesting feature of Equation (5) is that all sparsity contours are, by construction, parallel to the Chinchilla compute optimal line (Hoffmann et al., 2022), which denotes ideal utilization of training FLOPs for fully dense models; this can be clearly observed in the plots as well. However, we note that the Chinchilla line does not necessarily correspond to the $S = 0$ case since non-zero sparsity may be optimal in this regime (this is the case for sparse-FLOPs).

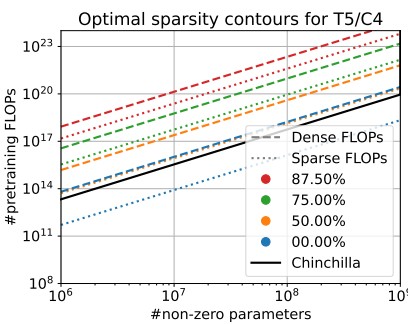

Figure 4: Optimal T5 sparsity contours.

The key take-away from these results is that as one trains significantly longer than Chinchilla (dense compute optimal), more and more sparse models start to become optimal in terms of loss for the same number of non-zero parameters. This is because the gains of further training dense models start to slow down significantly at some point, allowing sparse models to overtake them. We further illustrate this effect on a subset of our actual ViT data in Figure 5.

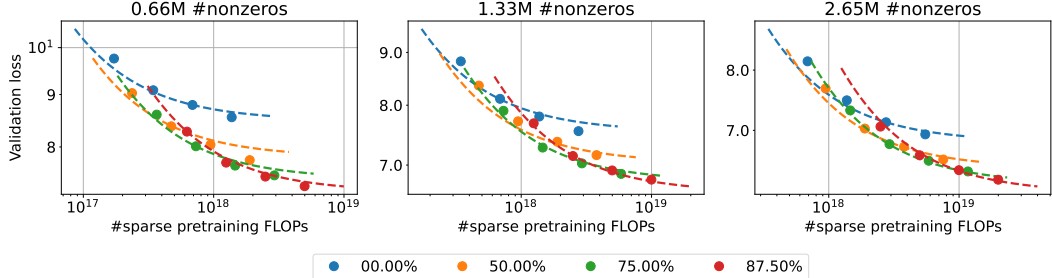

Figure 5: Loss vs. sparse pretraining FLOPs for ViT models of varying sparsity.

The practical question now is how much longer training is necessary? In terms of sparse FLOPs, 50% sparsity is already optimal for $< 2\times$ (ViT) and $< 3\times$ (T5) longer training than Chinchilla; for dense FLOPs it is $\approx 5\times$ and $\approx 70\times$, respectively. While the latter number seems quite high at first glance, we note that language models of the sizes we consider here are already typically trained for $> 100\times$ longer than Chinchilla (Brown et al., 2020). Additionally, larger models are being trained with more and more data as well, e.g., Llama2-7B with $\approx 14\times$ Chinchilla (Touvron et al., 2023b). In general, the optimal sparsity at a given point $(N, C)$ is lower for dense than sparse FLOPs since the former assumes that sparsity provides no benefits *during* training.

### 2.2.1 LIMIT PERFORMANCE

In the previous section, we have focused only on *when* sparse models become optimal but not *how much better* they can be compared to dense models. In this section, we study the following question: How much larger, and thus computationally more expensive, does a dense model need to be in order to match the loss of a smaller sparse model with very long training? Since we have found the scaling term in $D$ to not interact with sparsity in Section 2.1, it suffices to compute the increase in $N$ required to lower the loss by the same factor as the increase in $S$ via:

$$\text{gain}(S) = \Big(\frac{a_S(1 - S)^{b_S} + c_S}{a_S + c_S}\Big)^{-1/b_N}. \tag{7}$$

The gains for our particular scaling coefficients are shown in Table 1 (Left). They are to be interpreted in the following way: for example, a 75% sparse ViT with $N$ non-zeros will perform similar to a dense one with $\approx 2.17N$ parameters, when both are trained with *the same amount of data*. Crucially, this holds for *any* amount of data and thus also in the infinite limit when training is purely capacity bound. Hence, this expresses an equivalence between dense capacity and sparse capacity. We find that sparsity gains are very similar across vision and text domains, with the sweet-spot being around 75% sparsity at $\approx 2.15\times$ gain. This is in alignment with the view of sparsity as a

general increase in modeling power; we note that the gain is defined *relatively* to domain-specific improvements through increases in model size.

| Family | 0.500 | 0.750 | 0.875 |
|--------|-------|-------|-------|
| ViT/JFT | 1.60× | 2.17× | 2.63× |
| T5/C4 | 1.59× | 2.16× | 2.63× |

| Pattern | 0.50 | 0.75 |
|---------|------|------|
| n:4 | 1.56× | 1.62× |
| n:8 | 1.67× | 1.81× |

Table 1: (Left) Equivalent dense size multiplier to match performance of a sparse model. (Right) Dense size multipliers for n:m sparsity on T5/C4.

## 3 EXTENSIONS

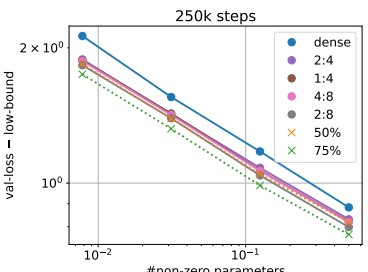

Figure 6: Loss vs. size plot for a subset of T5/C4 n:m sparsity data.

**N:M sparsity.** Complementing the *unstructured* sparsity exploration, we now also consider *structured* n:m sparsity, which can be accelerated on hardware (Pool & Yu, 2021; Hubara et al., 2021). Similar to how minor changes in the process (optimizer, model shape) generally only affect the multiplicative constants in dense scaling laws (Kaplan et al., 2020), we also expect minor changes in the sparsification process to only affect the sparsity term in (3). This can be exploited to fit laws based on significantly less runs: if the dense base scaling is known, one only has to fit $a_S$, $b_S$ and $c_S$ (just 3 rather than 7 parameters) to find the corresponding $L(S, N, D)$. We utilize this in the context of n:m sparsity by fitting new laws for 2:4 and 1:4 as well as 4:8 and 2:8 patterns, respectively, based only on a subset of our full grid in Appendix A. Concretely, we execute all runs involving either the least amount of steps, or the smallest model.

Figure 6 visualizes a subset of the collected data, displaying a very similar form to 2, which indicates that the general scaling law shape also holds for n:m sparsity. We also fit scaling laws (with Huber $\delta = 0.01$ as 0.75 patterns will otherwise be treated as an outlier) and calculate sparsity gains as in Section 2.2.1; see Table 1 (Right). In general, 2:4 and 4:8 perform both very similar to 50% (see Table 1 and also Figure 6), although the n:m estimates are slightly more noisy due to less data used in fitting the curves. Meanwhile, 1:4 brings almost no advantage and 2:8 only a slight improvement, which is contrary to our unstructured results. We suspect that the 75% patterns may simply be too stringent to significantly increase capacity beyond their 50% variants.

**Decoder-only models.** While we have focused on encoder-decoder models for language to cover a wide range of architectures, we now also validate our sparsity scaling laws in the context of standard decoder-only Transformers trained for auto-regressive language modeling. We follow the same setup as in our T5/C4 experiments, changing only model architecture and loss function. We again execute the experiment grid defined in Table 3 for 250K and 500K training steps, as well as a subset of the more expensive 1M step runs. Figure 7 demonstrates that all key sparse scaling properties still hold.

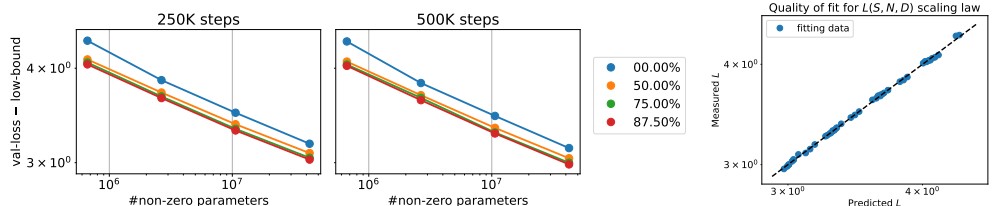

Figure 7: Visualization of decoder-only experiment and corresponding scaling fit quality.

**Pruning pretrained models.** Lastly, we consider a practical scenario where a set of existing *very well-trained* dense models should be made more efficient via pruning, using *a small fraction* of the compute spent for the initial pretraining. Our main interest here is to compare the efficiency of sparsifying from scratch and sparsifying from a pretrained checkpoint. For that purpose, we train ViT S/16, M/16 and B/16 models for 4 full epochs on JFT ( i.e., 16 billion images) and then start the same gradual sparsification procedure we used before from these checkpoints, for 5.6% of the pretraining budget (as the model is already pretrained, we start to sparsify immediately rather than after 25% of training). Finally, we use our scaling laws from Section 2.1 to determine the

amount of training necessary to produce equivalent models of the same quality when starting from scratch. Table 2 shows how much more/less data is required to achieve equivalent performance for sparsifying from scratch, when excluding/including the pretraining cost, respectively.

| Model | 0.500 | | 0.750 | | 0.875 | |
|---|---|---|---|---|---|---|
| | exc. | inc. | exc. | inc. | exc. | inc. |
| S/16 | $4.90\times$ | $0.25\times$ | $4.27\times$ | $0.23\times$ | $2.45\times$ | $0.13\times$ |
| M/16 | $4.76\times$ | $0.25\times$ | $4.18\times$ | $0.22\times$ | $2.57\times$ | $0.14\times$ |
| B/16 | $4.35\times$ | $0.23\times$ | $4.00\times$ | $0.21\times$ | $2.72\times$ | $0.14\times$ |

Table 2: Relative amount of data required for sparsifying from scratch to match the validation loss of pruning from a pretrained model, when pretraining cost is excluded (exc.) and included (inc.).

If the model already exists and there is thus no pretraining cost, then starting from such a checkpoint is $> 4\times$ more efficient then sparsifying from scratch for 0.5/0.75, and $> 2\times$ for 0.875 sparsity, respectively. The reason why the efficiency gains are decreasing with higher sparsity is most likely the increased divergence from the initial starting point. At the same time, when the pretraining cost is counted as well, pruning throughout the whole training process appears to be $\geq 4\times$ more efficient, relative to the $\approx 5\%$ pruning of pretraining budget. Overall, these results clearly demonstrate that, while the sparsification process benefits significantly from a better trained initial model, it only does so to a limited extent.

## 4 RELATED WORK

**Sparsity & pruning.** Sparsity has a long history (LeCun et al., 1989; Hassibi et al., 1993) and a large number of works have been published on this topic (Hoefler et al., 2021). State-of-the-art methods range from simple gradual removal of the smallest weights (Zhu & Gupta, 2017), to partial or full sparse training (Mocanu et al., 2018; Jayakumar et al., 2021; Peste et al., 2021), approximate Hessian-based metrics (Singh & Alistarh, 2020; Frantar et al., 2021; Kuznedelev et al., 2023a) and "soft" sparse optimization (Kusupati et al., 2020; Sanh et al., 2020). Sparsity can lead to substantial practical speedups with specialized inference algorithms (Kurtz et al., 2020; Elsen et al., 2020). Yet, most of those works focus on relatively simple tasks like ResNet50/ImageNet or BERT/GLUE.

In contrast, much less is known when it comes to sparsifying Transformers trained on massive datasets: The Appendix of Gopher (Rae et al., 2021) conducts pruning experiments for a generative language modelling task and finds that, when trained for the same amount of steps, sparse models can outperform dense ones, but leaves open whether this is also possible when accounting for the significantly increased compute spent for producing those sparse models, relative to equivalently sized dense ones. Similarly, Cerebras (2022) prunes a GPT-like model, also using significantly more data than its dense baseline. Recently, SparseGPT (Frantar & Alistarh, 2023) showed that it is possible to impose weight-sparsity on extremely large language models, even without retraining; yet, it remains unclear if this can also be done on smaller networks that are trained on more data.

**Scaling laws.** The key behind the success of Transformers is their exceptional scaling: increasing model size and/or data brings consistent performance improvements, even at huge scale. Further, this scaling behavior is predictable, following simple power-law curves (Kaplan et al., 2020). This can be utilized to construct a family of compute optimal models (Hoffmann et al., 2022). More recently, scaling laws are being extended to more specialized applications, e.g.: optimizing model shapes (Alabdulmohsin et al., 2023), routing mechanisms (Clark et al., 2022), repeating training data multiple times (Muennighoff et al., 2023) and several downstream tasks (Caballero et al., 2023). However, not much is known about the scaling of weight sparsity for such models.

Rosenfeld et al. (2021) studies the relationship between width, depth and weight density for pruning pretrained ResNets trained primarily on the CIFAR dataset (Krizhevsky et al., 2009), which is nowadays considered very small. Contrarily, we consider modern Transformers trained on datasets many orders of magnitude larger and focus particularly on the data/compute dimension that is crucial in this context, but not relevant in the setting of Rosenfeld et al. (2021).

**Transformer efficiency.** Making (large) Transformers more efficient is currently a highly active area of research. Probably the currently most popular and practical approach is quantization, that is reducing the numerical precision (Frantar et al., 2022; Dettmers & Zettlemoyer, 2023; Xiao et al., 2022). Further, there are also many works on Mixture-of-Expert (MoE) models, which bound

the overall computation cost per sample (Du et al., 2022; Fedus et al., 2022; Artetxe et al., 2022; Riquelme et al., 2021). MoEs are a form of *dynamic activation* sparsity, which is very different from the *static weight* sparsity that we study; the former trades off increased memory for faster inference, whereas the latter reduces both inference *and* memory costs. In general, quantization, MoEs and weight sparsity are complementary and may be stacked for compound gains (Kurtic et al., 2022).

## 5 DISCUSSION

**Limitations.** While our study is based on extensive experiments across models and domains, it also has limitations, which we discuss here and plan to address in future work. First, our sparsification recipe was optimized for robustness and scalability across a wide range of setups, rather than to fully maximize sparsity versus accuracy in a particular setting. We believe that specific coefficient values can be improved with extensive setting-specific tuning and better sparsification techniques; however, the general nature of our scaling law will remain consistent.

We focus on settings where pruning is applied to pretraining tasks with massive amounts of data and compute. This is ideal in terms of usability, as down-stream (finetuning) applications directly benefit from the pruned model, but it also makes compression quite challenging. We think higher sparsity rates can be achieved if pruning is applied directly for specialized applications that only require a subset of the model's capabilities. Similarly, our study focuses on the infinite data setting, which essentially eliminates overfitting, as only a single pass over the data is performed. Sparsity could be particularly effective when data is limited and thus multiple epochs are performed.

Our proposed sparse scaling law suggests that higher sparsity is always better (but with potentially quite quickly saturating improvements), which may not be true in extremes. For very high sparsity (e.g., $64\times$ compression) we sometimes see slightly worse performance, presumably due to imperfections in the pruning and optimization process. This phenomenon could potentially be modelled by a quadratic, but the present study treats it as a bottleneck-case that is not necessarily captured.

Finally, the main goal of this study is understanding core scaling relationships. Thus, we focused on the most fundamental cost metric, non-zero parameter count. However, in practice, sparsity acceleration can be complex: current software/hardware may not provide ideal speedups and models generally also contain operations (e.g., layer-norms, attention) which do not benefit from weight sparsity. Extending our scaling results to more target metrics is an interesting topic for future work.

**Compatibility with other works.** We will now briefly discuss how our scaling insights line up with existing sparsification results on similar models/datasets. First, the results in the Appendix of Rae et al. (2021), for a decoder-only text-generation model, are consistent with our scaling laws; the improvement through sparsity appears to be similar for each model size and their maximum size advantage of $2.5\times$ observed at $0.9$ sparsity is quite close to our limit gains in Section 2.2.1.

In contrast, Cerebras (2022) report a significantly better gain of $\approx 5\times$, but in a quite different setting where the baseline is training (not inference) compute optimal and sparsification uses $> 5\times$ more data than the dense comparison point. This is not inconsistent to our results: if we query our fitted T5 scaling law (see Section 2.1) with this setup, we predict 1.54 loss (dense 1B params, 20B tokens) vs. 1.48 loss (80% sparse & 200M non-zeros, 100B tokens), in favor of the sparse model.

Finally, SparseGPT (Frantar & Alistarh, 2023) notes that post-training pruning becomes easier as the model size increases. However, they do not perform any retraining, and observe this effect primarily relative to the respective unpruned base model, not in terms of improvements over the Pareto size-vs-loss frontier that we focus on. Hence, we believe that this is likely more related to the pretrained models' initial robustness to pertubations rather than the architecture's inherent sparsifiability.

**Practical implications.** Our scaling insights lead to a number of practical consequences: Sparsity seems to affect each model size in approximately the same way, while remaining mostly independent of the amount of training data used. This provides evidence that good pruning performance in less expensive settings should generalize to performance at scale, which will hopefully accelerate research on new sparsification recipes and algorithms. Additionally, we have shown that optimal sparsity levels continuously increase with longer training. Sparsity thus provides a means to further improve model performance for a fixed final parameter cost. In particular, when training beyond Chinchilla optimality, where simple dense training starts to run into diminishing returns, sparsity can provide a clear alternative. Thus, our findings can be interpreted as providing practical motivation for further developing sparsity support.

## 6 ETHICS STATEMENT

The primary goal of this research is to examine the interaction between parameter sparsity and the scaling behavior of foundation models, and as such it does not pose immediate ethical concerns. The techniques explored focus on compression and computational efficiency, and thus will lead to no harm to individuals or potential bias towards users. The recommendations provided for leveraging sparsity aim to advance technology and should serve to improve the computational burden caused by large models. As such, they should lead to positive societal outcomes.

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

# A  EXPERIMENTAL SETUP

This section discusses our experimental choices and hyper-parameters, as well as technical details for sparsity-aware AdaFactor and iterative n:m pruning.

**Models & datasets.**    We consider two standard deep learning applications: vision and language. For the former, we focus on Vision Transformers (Dosovitskiy et al., 2021) trained for multi-label image classification on the JFT-4B dataset (Dehghani et al., 2023), consisting of 4 billion images; for the latter, we consider encoder-decoder T5 models (Raffel et al., 2020b) (improved 1.1 version (Google, 2023b)) trained for masked-language-modelling on C4 (Raffel et al., 2020b), consisting of 150+ billion tokens. These choices allow us to study the generality of our laws not just across vision and language but also for different kind of pretraining objectives and variations of Transformer architectures.

**Training hyper-parameters.**    For the most part, we reuse the optimized training hyper-parameters of the original ViT-scaling (Zhai et al., 2022) and T5 paper (Raffel et al., 2020b), respectively. Our only notable change is that we do not factor the second moment of the respective AdaFactor-based (Shazeer & Stern, 2018) optimizers (however, we still apply relative learning rate scaling and RMS clipping for T5); this is done since factorized moments for pruning and sparse optimization are not yet very well studied. Further, we train T5 models with batchsize 128 (similar to most ablation studies in the original paper (Raffel et al., 2020b)) in order to perform sufficiently many optimization steps also for experiments with lower total amounts of training data, which we found important to obtain stable sparse results through model and data scaling.

**Model sizes.**    When it comes to selecting our particular model dimensions, two things must be taken into account: (a) we are particularly interested in the inference-optimal *overtraining regime* (Touvron et al., 2023a) where models get close to their capacity limit, and (b) to produce a model with $N$ non-zeros and sparsity $S$, we actually need to train a model that is $1/(1 - S)$ times larger than a dense model of size $N$. In combination with the fact that we need to repeat the entire training sweep for multiple sparsity levels, this limits the size of models we can study while keeping compute requirements feasible. Specifically, we start from the base variants of both ViT and T5 (B/16 and t5-base). Then we generate models of appropriate sizes by scaling only the Transformer's hidden dimension and keeping all other shape parameters constant. This way we can get quite precise size-matches between models and sparsities, facilitating direct comparisons (not all default family models are exactly $2\times$ steps apart and a 50% sparse model would thus not always be directly comparable to the next smallest dense variant); we did not observe any notable performance decrease for dense models using this scaling strategy, at the sizes we study.

**Sparsity configurations.**    We focus primarily on the most fundamental sparsity type, *unstructured* sparsity, but also perform some investigations for the more practical *n:m* pruning pattern (Zhou et al., 2021; Pool & Yu, 2021) where only $n$ out of $m$ consecutive weights are non-zero. We uniformly sparsify all linear layers in the Transformer backbone, which effectively avoids layer collapse (Wang et al., 2020), or other edge cases that may otherwise occur in our sweeps, and generally works decently well for Transformer models. On T5 models, we also sparsify the rather large embeddings to the amount necessary for parameter matching a smaller dense version.

Preliminary experiments indicated quickly diminishing returns for very sparse models, which, as discussed previously, are in addition quite expensive to train. Thus, we focus on three medium sparsities: 50%, 75%, 87.5%, corresponding to a $2\times$, $4\times$ and $8\times$ compression rate, respectively. Our implementation is based on the recently proposed Jaxpruner library (Lee et al., 2023), which is easy to integrate into the offical ViT (Google, 2023a) and T5 (Google, 2023b) codebases.

**Pruning strategy.**    As we intend to execute substantial sweeps across model size, training data and sparsity, it is essential that our pruning method is highly robust and does not require hyper-parameter retuning for each run. A natural candidate is gradual magnitude pruning (GMP) (Zhu & Gupta, 2017), which is well studied and known to be very reliable. At the same time, GMP is usually quite competitive with more complex techniques (Singh & Alistarh, 2020; Kurtic & Alistarh, 2022), especially at the medium sparsity levels that we focus on. We also tested a variation of GMP which incorporates diagonal second-order information (Kurtic et al., 2022), but found it to perform almost identically in our setting. Further, we tried AC/DC (Peste et al., 2021), STE (Lin et al.,

2020) and RigL (Evci et al., 2020) (which achieve strong results on classic benchmarks) but saw similar or worse performance, while being more sensitive to hyper-parameters as we scale (see also Appendix B). Thus, we ultimately decided to use GMP.

In terms of specific hyper-parameters, we prune using a cubic schedule starting after 25% of training and ending at 75%, updating every 100 steps. Our sparsification interval was chosen so that pruning begins with a reasonably well trained model and ends with sufficient finetuning of the final sparse structure. However, we performed ablations for frequency/start/end in Appendix B and did not find the process to be too sensitive to those hyper-parameters (except for when the pruning interval is really short).

**Sweep grids.** Table 3 lists the grid parameters that we sweep over. For ViTs, we consider 7 target models sizes in $2\times$ increments each, while we use 4 targets sizes in increments of $4\times$ for T5. Vision Transformers are trained for 4 different lengths, with the longest corresponding to $\approx 1.8$ billion images; language models are trained for 3 different lengths up to $\approx 65$ billion tokens. The set of sparsity targets is the same in both cases, corresponding to 2, 4 and $8\times$ compression rate. Overall, the ViT grid was designed to be more extensive whereas the T5 setup was chosen to be more efficient.

| Model family | ViT | T5 |
|---|---|---|
| #Non-zero params | 0.66M, 1.33M, ..., 42.4M | 1.3M, 5.3M, ..., 85M |
| Training steps | 55K, 110K, 220K, 440K | 250K, 500K, 1M |
| Sparsities | 0.0, 0.5, 0.75, 0.875 | 0.0, 0.5, 0.75, 0.875 |
| Total #runs | 112 | 48 |

Table 3: Grid definition for our main scaling sweeps.

### A.1 TECHNICAL DETAILS

**Sparsity-aware RMS.** AdaFactor (Shazeer & Stern, 2018) as employed by T5 (Raffel et al., 2020b) defines the learning rate relatively, scaling it by the root-mean-square (RMS) of each weight tensor, respectively. We find that this does not interact well with high sparsity, as a tensor with many zeros tends to have a lower RMS, resulting in a smaller learning rate, which is especially problematic as high levels of sparsification require more recovery during the pruning process. To work around this, we always calculate the RMS only over *unpruned* weights, which effectively alleviates this problem. We also apply the same technique to AdaFactor's RMS clipping threshold, but note that this is much less critical than the learning rate scaling.

**Iterative n:m pruning.** Sparsifying to the n:m pattern is usually done by pruning in one-shot, followed by finetuning (Zhou et al., 2021), or directly training with a dynamic pruning mask via straight-through gradient estimation (Lu et al., 2023). We take a different approach in this work: we gradually remove the smallest weights while ensuring that at least $n$ weights remain in each group of size $m$. This effectively generalizes the highly robust gradual pruning paradigm to the n:m setting. Not only does *gradual n:m pruning* unify our setups between unstructured and structured sparsity experiments, we also found it to work reliably across scales with the same hyper-parameters, a highly useful property for scaling studies. A simple and efficient implementation of this scheme is shown in Algorithm 1: the key is to temporarily set the largest $n$ items in each group to $\infty$, thus ensuring that they are always picked by an unstructured topk selection.

---

**Algorithm 1** Prune weights $\mathbf{w}$ to sparsity $s \leq 1 - n/m$ where each group of $m$ weights contains at most $n$ zeros.

---

$I_{\text{nm}} \leftarrow \text{topk-nm}(|\mathbf{w}|, n, m)$
$\mathbf{w}' \leftarrow \text{copy of } \mathbf{w}$
$\mathbf{w}'_{I_{\text{nm}}} \leftarrow \infty$
$I_{\text{unstr}} \leftarrow \text{topk-unstr}(|\mathbf{w}'|, 1 - s)$
$\mathbf{w}_{-I_{\text{unstr}}} \leftarrow 0$

## B    PRUNING ABLATIONS

We ablate gradual magnitude pruning hyper-parameters by sparsifying ViT-B/16 for 900M images to $S = 0.9375$ sparsity. A high $S$ was chosen in order to amplify differences between parameter settings; we vary the (relative) start and end point of gradual pruning as well as the update frequency of the mask; the pruning schedule is always cubic. As can be seen in Table 4, most configurations perform very similar, except when the pruning period is too short overall. We ultimately pick the 25-75/100 setup to ensure that there is sufficient time for training a decent model before starting pruning, as well as for properly finetuning the final sparse version, which we think could be helpful for some points in our main scaling experiment grid.

| start | end | freq | accuracy |
|-------|-------|------|----------|
| 0.250 | 0.750 | 100 | 45.28 |
| 0.250 | 0.750 | 50 | 45.08 |
| 0.250 | 0.750 | 200 | 45.13 |
| 0.125 | 0.875 | 100 | 45.33 |
| 0.475 | 0.625 | 100 | 44.65 |
| 0.125 | 0.625 | 100 | 45.13 |
| 0.375 | 0.875 | 100 | 45.04 |

Table 4: Ablation study of gradual magnitude pruning hyper-parameters.

**AC/DC.**    We also experimented with the AC/DC method (Peste et al., 2021), a sparse training approach that was recently shown to yield very strong results on standard (non-foundation model) benchmarks Kuznedelev et al. (2023b). We use a sparse and dense cycle length of 20K steps (10K each phase) and apply AC/DC only during the same pruning period as our GMP setup to ensure the same pre- and post-sparsification finetuning. On smaller T5 models, AC/DC works well but yields very similar results to GMP. On larger models, however, AC/DC appears to require some hyper-parameter reconfiguration for higher sparsities. Since per-model hyper-parameter tuning is inconvenient for large scaling sweeps, while initial results also did not suggest clear improvements, we stuck to well established GMP. In general, we note that even for classic benchmarks, major differences between pruning methods tend to appear mostly at very high sparsities (Singh & Alistarh, 2020). Nevertheless, we think that more extensively investigating advanced sparsification approaches in the context of massive pretraining datasets is an interesting topic for future work.

| #nnz | 0.500 | | 0.750 | | 0.875 | |
|------|-------|-------|-------|-------|-------|-------|
|      | GMP | AC/DC | GMP | AC/DC | GMP | AC/DC |
| 1.3M | 17.11 | 17.11 | 15.64 | 15.96 | 14.73 | 14.59 |
| 42M | 6.11 | 6.11 | 5.87 | 6.05 | 5.81 | 6.11 |

Table 5: Comparing validation perplexity of T5 models trained for 250K steps using GMP and AC/DC, respectively; we show perplexity as losses at these levels should be compared in log-scale.

## C    SCALING COEFFICIENTS

Table 6 lists the fitted coefficient values for the scaling results presented in the main paper; $D$ is assumed to be the number of images for ViT and the number tokens for T5. The fitting errors are also shown, where we note again that they correspond to the Huber-loss with $\delta = 0.01$ for ViT/JFT and the Huber-loss of $\log L$ with $\delta = 0.001$ for T5/C4, following (Hoffmann et al., 2022).

For n:m sparsity, we only refit the sparsity coefficients $a_S$, $b_S$ and $c_S$, preserving the other values from the corresponding unstructured results. While the fitting procedure may not be guaranteed to be convex, we find the process to converge to virtually the same values from different random starting points.

| Model | Sparse | $a_S$ | $b_S$ | $c_S$ | $b_N$ | $a_D$ | $b_D$ | $c$ | Error |
|-------|--------|-------|-------|-------|-------|-------|-------|-----|-------|
| ViT/JFT | unstr. | 2.94e+2 | 0.821 | 4.68e+2 | 0.392 | 2.37e+8 | 0.890 | 4.517 | 4.93e-4 |
| T5/C4 | unstr. | 1.68e+1 | 0.722 | 4.50e+1 | 0.245 | 6.90e+8 | 0.203 | 0.651 | 7.60e-6 |
| T5/C4 | n:m | 8.64e+1 | 2.752 | 5.36e+2 | – | – | – | – | 2.1e-5 |

Table 6: Fitted coefficients of the scaling laws presented in the main paper.

## D  OPTIMAL SPARSITY DERIVATIONS

**Sparse costs.**  We now discuss how to derive the sparse cost factor $c_{\text{mul}}(S)$ given by Equation 4 in Section 2.2 for our particular pruning setup. For the first 25% of training, we use a dense model that is $1/(1-S)$ times larger, incurring cost $0.25/(1-S)$, while for the last 25% we are training a model with sparsity $S$ of the same cost as the dense reference model, thus contributing a $0.25$ term. For the middle 50%, we prune in the cubic $S - S \cdot (1-t)^3$ schedule (Zhu & Gupta, 2017), where $t \in [0,1]$. The cost spent over this period is given by 1 minus (we care about density rather than sparsity) the integral over the full range of $t$, which is $(1 - 0.75 \cdot S)$, further multiplied by 0.5 to cover the duration.

**Contour lines.**  To determine the contours of $S_{\text{opt}}(N, C)$, i.e., the regions where a particular sparsity $S$ is optimal, we first query $L(S, N, D)$ with $D_S = D/c_{\text{mul}}(S)$, where $D = C/6N$ is the amount of data corresponding to dense training FLOPs $C$ relative to which $c_{\text{mul}}$ is defined, giving:

$$(a_S(1-S)^{b_S} + c_S) \cdot \left(\frac{1}{N}\right)^{b_N} + \left(c_{\text{mul}}(S) \cdot \frac{a_D}{D}\right)^{b_D} + c. \tag{8}$$

Next, we optimize for $S$ by differentiating and setting the result equal to 0:

$$-a_s b_s(1-S)^{b_S-1} \cdot \left(\frac{1}{N}\right)^{b_N} + b_D\left(c_{\text{mul}}(S) \cdot \frac{a_D}{D_S}\right)^{b_D-1} \cdot \left(c'_{\text{mul}}(S) \cdot \frac{a_D}{D_S}\right) = 0. \tag{9}$$

Finally, the form presented in the main paper is constructed by a few algebraic simplifications:

$$b_D\left(c_{\text{mul}}(S) \cdot \frac{a_D}{D_S}\right)^{b_D-1} \cdot \left(c'_{\text{mul}}(S) \cdot \frac{a_D}{D_S}\right) = a_s b_s(1-S)^{b_S-1} \cdot N^{-b_N} \tag{10}$$

$$b_D \cdot \frac{c'_{\text{mul}}(S)}{c_{\text{mul}}(S)} \cdot \left(c_{\text{mul}}(S) \cdot \frac{a_D}{D_S}\right)^{b_D} = a_s b_s(1-S)^{b_S-1} \cdot N^{-b_N}. \tag{11}$$

**Closed form solution.**  The closed form solution for $S_{\text{opt}}(N, C)$ assuming dense costs $c_{\text{mul}}(S) = 1/(1-S)$ can be found by solving the above contour equation for $S$:

$$\left(\frac{1}{1-S}\right)^{b_D+1} \cdot b_D\left(\frac{a_D}{D_S}\right)^{b_D} = (1-S)^{b_S-1} \cdot a_s b_s \cdot N^{-b_N} \tag{12}$$

$$\left(\frac{1}{1-S}\right)^{b_D+b_S} = \frac{a_S b_S}{b_D} \cdot N^{-b_N}\left(\frac{a_D}{D_S}\right)^{-b_D} \tag{13}$$

$$(b_D + b_S) \cdot \log\left(\frac{1}{1-S}\right) = \log\left[\frac{a_S b_S}{b_D} \cdot N^{-b_N}\left(\frac{a_D}{D_S}\right)^{-b_D}\right] \tag{14}$$

$$\frac{1}{1-S} = \exp\left\{\log\left[\frac{a_S b_S}{b_D} \cdot N^{-b_N}\left(\frac{a_D}{D_S}\right)^{-b_D}\right]/(b_D + b_S)\right\} \tag{15}$$

$$S = 1 - \exp\left\{-\log\left[\frac{a_S b_S}{b_D} \cdot N^{-b_N}\left(\frac{a_D}{D_S}\right)^{-b_D}\right]/(b_D + b_S)\right\}. \tag{16}$$

Further algebraic manipulations yield:

$$S = 1 - \exp\left\{\log\left[\frac{b_D}{a_S b_S} \cdot N^{b_N}\left(\frac{a_D}{D_S}\right)^{b_D}\right]/(b_D + b_S)\right\} \tag{17}$$

$$S = 1 - \exp\left[\left(\log\frac{b_D}{a_S b_S} + b_N \log N + b_D \log\frac{a_D}{D_S}\right)/(b_D + b_S)\right]. \tag{18}$$

$$S = 1 - \exp\left[\left(\log\frac{b_D}{a_S b_S} + b_N \log N\right)/(b_D + b_S)\right] \cdot \left(\frac{a_D}{D_S}\right)^{b_D/(b_D+b_S)}. \tag{19}$$

Substituting $D_S$ by its expression in $C$ and guaranteeing that the result is always non-negative (since sparsity $< 0$ does not exist) brings the formula given in Section 2.2.

# E    IMPACT OF SPARSITY ON DOWNSTREAM TASKS

As discussed in the main text, the focus of this study, similar to most other works on scaling laws, lies on modeling the *pretraining task validation loss*. For dense models it is known, that this loss is strongly correlated with performance on downstream (e.g. few-shot) applications (Hoffmann et al., 2022). This effect has also been observed for sparse models, for example in the context of transfer learning for ImageNet models (Iofinova et al., 2022). We now provide additional evidence, for our particular sparse models by evaluating them on a suite of common few-shot tasks.

Concretely, we consider all dense and sparse models from our main sweep and plot few-shot accuracy vs. pretraining loss; the results are shown in Figure 8. As can be seen, sparse models exhibit very similar behavior to dense ones, both in terms of accuracy achieved for a fixed validation loss as well in terms of overall noise level that is typical for each task. This confirms that sparse models with better pretraining validation loss, like dense models, indeed also perform, on average, better on few-shot tasks.

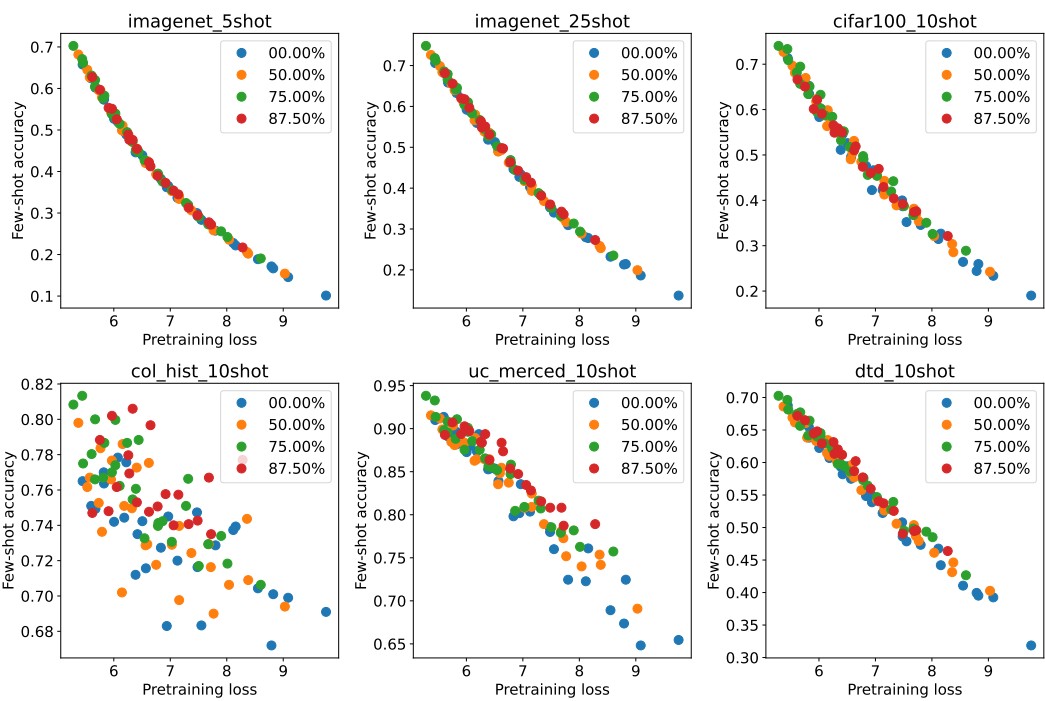

Figure 8: Few-shot accuracy vs. pretraining loss for dense and sparse ViT/JFT models.

# F    PRACTICAL SPARSITY ACCELERATION

Although we focus primarily on developing a general understanding of sparsity scaling, rather than on accelerating sparse model in practice, there are many works that study the latter. We now summarize some of them.

Network weights that are exactly zero must not be stored and can be safely skipped during computation. This means that, in principle, sparse models require both less memory as well as less compute to run than their dense counterparts. For CPU inference, this compute reduction can be effectively translated into end-to-end speedups using sparsity-aware matrix algorithms (Elsen et al., 2020; Kurtz et al., 2020). Custom hardware can also achieve strong throughput improvements for sparse models: for example, Thangarasa et al. (2023) report almost $4\times$ speedup on a matrix multiplication with 75% sparse weights. On modern GPUs, effectively utilizing fully unstructured sparsity is challenging due to irregular computation patterns. However, newer generations of NVIDIA GPUs include dedicated hardware support for semi-structured n:m sparsity (Pool & Yu, 2021), yielding up to $2\times$ speedup over fully dense matrix multiplications.

While effectively utilizing weight sparsity *during* training is still an active of research, there are already some promising results. Emani et al. (2023) report substantial end-to-end training speedups for training large, unstructured sparse, Transformer models on custom hardware. Meanwhile, Nikdan et al. (2023) develop sparsity-aware backpropagation kernels for efficient CPU training.

Finally, we note a very recent line of work which uses unstructured sparsity (Xia et al., 2023; Kurtic et al., 2023) primarily for reducing model size, in order to fit massive Transformer models onto smaller accelerators. Further, for memory bound generative inference, this brings practical as well, by reducing transfer costs between GPU main memory and registers due.

