# OpenReview forum: "Scaling Laws for Sparsely-Connected Foundation Models"
_ICLR.cc/2024/Conference — ICLR 2024 spotlight_

### Official Review · Reviewer_Tdrh · 2023-10-26

**Soundness:** 3 good
**Presentation:** 2 fair
**Contribution:** 3 good
**Rating:** 6
**Confidence:** 3

**Summary:**

This work uses scaling law to describe the relationship between weight sparsity, number of non-zero parameters and amount of training data.

**Strengths:**

1, In-depth investigation of the relationship between validation loss and sparsity, number of non-zero parameters and training data. Useful for understanding sparsification better.

2, Derived optimal sparsity for a given inference size and training budget.

**Weaknesses:**

1, Experiments are only tested on two examples. Though due to limited resources, more examples may be challenging, two examples may not be very convincing.

2, The scaling law has too many degrees of freedom (7 free parameters) that could be hard to generalize to new dataset without sufficient information.

**Questions:**

1, What are the 7 free parameters fitted and how are these parameters changes for different dataset and different hyperparameters $N, D, S$?

2, Procedure to identify $S_{opt}$ in 2.2 Optimal Sparsity is confusing. Could you explain this procedure with more details?

---

> ### Author Response · Authors · 2023-11-17
> **Author Response**
>
> Thank you for the review! We respond to your comments and questions in detail below.
>
> > 1, Experiments are only tested on two examples. Though due to limited resources, more examples may be challenging, two examples may not be very convincing.
>
> Our main experiments feature 112 training runs for ViT models and 48 training runs for T5 models, plus several runs for n:m sparsity, pruning pretrained models, ablation studies and two larger extrapolation experiments. Many of these take days to execute on multiple accelerators (one extrapolation experiment for example takes 2 days on 128 devices).
>
> Additionally, towards addressing this concern, we **have conducted additional experiments for autoregressive decoder-only models**, showing that our scaling laws also hold in this context. We present those new results in Appendix G of the most recent revision or, for convenience, at this anonymized link: https://github.com/iclr24-1372/iclr24-1372/blob/master/decoder-only.pdf.
>
> Finally, we would like to note that most scaling works focus only on a single domain and model family, while we study both vision and text models (and now also encoder-decoder and decoder-only variants). This highlights the generality of our sparsity scaling laws.
>
> > 2, The scaling law has too many degrees of freedom (7 free parameters) that could be hard to generalize to new dataset without sufficient information.
>
> Standard dense scaling laws usually have at least 5 parameters for two inputs N (model size) and D (training data). Hence, we think allowing 2 more parameters to model the impact of an additional third input S (sparsity) is a reasonable increase. The 7 free parameters are fitted over 112 ViT and 48 T5 training runs, respectively, hence the #samples to #parameters ratio is ~16 and ~7, which limits overfitting capability. At the same time, we still observe good fits and extrapolation performance.
>
> We note that the coefficients of a scaling law are usually fit on a per model family and dataset/task basis, only the overall functional form of the law is supposed to generalize across settings, which we demonstrate by studying both vision and language models.
>
> > 1, What are the 7 free parameters fitted and how are these parameters changes for different dataset and different hyperparameters S, N, D?
>
> First, we would like to note that $S$/$N$/$D$ are inputs to our scaling law and not hyper-parameters, the same coefficient values ($a_S$, $b_S$, …) are used to model validation losses for all $S$/$N$/$D$ configurations. Different model family and dataset/task combinations can have coefficients that are quite different; we list the particular values we obtained for our experiments in Appendix C. However, the overall functional form of the scaling law remains consistent across domains (vision & text) and architectures (encoder-only, encoder-decoder).
>
> >2, Procedure to identify in 2.2 Optimal Sparsity is confusing. Could you explain this procedure with more details?
>
> The high level idea is as follows:
> 1. We first reparametrize our $L(S, N, D)$ scaling law in terms of $L(S, N, C)$ by replacing $D$ with the corresponding amount of training data that, for a model with parameter count $N$ and sparsity $S$, leads to overall compute $C$.
> 2. Next, we would like to minimize $L(S, N, C)$ with respect to $S$, in order to determine the best sparsity value. This is done in standard fashion by setting the derivative of $L$ with respect to $S$ to 0.
> 3. Our expression given for the contour lines is found by rearranging terms on both sides. Similarly, $S_\text{opt}$ is determined by solving for $S$ with standard algebraic manipulations. We provide those algebraic details in Appendix D.

---

### Official Review · Reviewer_1amU · 2023-10-30

**Soundness:** 3 good
**Presentation:** 3 good
**Contribution:** 2 fair
**Rating:** 6
**Confidence:** 5

**Summary:**

This paper studied the scaling laws for sparse foundation models. Specifically, they built a scaling law for how sparse models would perform as a function of sparsity, model size and pre-training data size. The authors evaluated on ViT and T5 architectures and showed that the constructed scaling laws can be used to easily predict the optimal sparsity without manually re-training the models.

**Strengths:**

1. The experimental results are comprehensive. I appreciate that the authors include experiments on hardware friendly structured sparsity.
2. Deriving scaling laws for sparse foundation models is important. As foundation models require huge compute to train, understanding their scaling behavior and relationship to sparsity is crucial for predicting model performance.
3. The empirical results show that ViT and T5 show strong scaling curves for sparsity and their performances can be well predicted from the constructed scaling law.

**Weaknesses:**

1. My main concern lies in the contribution of section 2 on fair evaluation, which argued that a sparse network should be compared with a dense model with the same number of parameters and compute. A very similar argument is made in [1] (See “training budget” in section 3), which argued that the compute budget for comparing sparse and dense models should be the same. It would be good to clarify the differences to [1] in the paper.
2. The evaluation on the encoder-decoder architecture T5 seems limited. The paper do not evaluate the more popular autoregressive decoder-only architecture but the original scaling laws is built on such autoregressive language models.

[1] Rethinking the value of network pruning. ICLR 2019.

**Questions:**

1. I am curious as to whether lottery ticket hypothesis would hold as to foundation models, as it is the most impactful pruning paper in recent years. Since this work is closely related, i am wondering if the authors have explored the LTH setting for sparse training. If not, it would be good to add a discussion on LTH [2].
2. Can the proposed scaling laws be useful in the extreme sparsity regime, e.g. 95% or even 99%? Or is it applicable only for the sparsity level within a range?

[2] The Lottery ticket hypothesis: Finding Sparse, Trainable Neural Networks. ICLR 2019.

---

> ### Author Response · Authors · 2023-11-17
> **Author Response**
>
> Thank you for the review! We respond to your comments and questions in detail below.
>
> > My main concern lies in the contribution of section 2 on fair evaluation, which argued that a sparse network should be compared with a dense model with the same number of parameters and compute. A very similar argument is made in [1] (See “training budget” in section 3), which argued that the compute budget for comparing sparse and dense models should be the same. It would be good to clarify the differences to [1] in the paper.
>
> The main purpose of this paragraph was to highlight that this resource normalization is critical when dealing with foundation models that exhibit strong scaling behavior. This is not as important on standard smaller-scale pruning benchmarks (e.g., CIFAR/ImageNet or BERT/GLUE) and is thus often neglected. Nevertheless, as pointed out by the reviewer, training budget normalization has indeed been considered by some prior works as well, in different contexts. We now explicitly state this in Section 2 of the newest revision, referencing Liu et al. (2019) [1], as well as Jin et al. (2022).
>
> Apart from the fact that those works also highlight similar considerations regarding fair evaluation of sparse models, they are quite different in scope. Specifically, neither of them develops scaling laws or studies foundation models.
>
> > The evaluation on the encoder-decoder architecture T5 seems limited. The paper do not evaluate the more popular autoregressive decoder-only architecture but the original scaling laws is built on such autoregressive language models.
>
> We initially selected the T5 encoder-decoder architecture to demonstrate that our sparsity scaling law is also applicable to slightly more complex Transformer variants. However, we agree that a standard autoregressive decoder-only setup would be a useful addition and **have conducted an additional scaling sweep for autoregressive decoder-only Transformers.**
>
> Specifically, we train several autoregressive decoder-only Transformers for next-token-prediction on the C4 dataset, using similar size/sparsity/data settings as in our T5 experiments. The results are presented in Appendix G of the most recent revision or, for convenience, at this anonymized link: https://github.com/iclr24-1372/iclr24-1372/blob/master/decoder-only.pdf.
>
> Overall, we observe all the key aspects of our scaling law also in this setting (sparsity acts as a constant multiplier on size scaling, is mostly independent of the amount of training data, and yields diminishing returns at higher values), which consequently leads to a good coefficient fit as well. This suggests that standard autoregressive decoder-only models exhibit the same general sparsity scaling behavior as ViTs and encoder-decoder models, further confirming our results.
>
> > I am curious as to whether lottery ticket hypothesis would hold as to foundation models, as it is the most impactful pruning paper in recent years. Since this work is closely related, i am wondering if the authors have explored the LTH setting for sparse training. If not, it would be good to add a discussion on LTH [2].
>
> We did not explore lottery tickets specifically; this is primarily because first training a dense model only to select a mask and then rewind any optimization progress (by resetting the weights to their initial values) is not a particularly efficient pruning strategy. This is especially the case when fairly accounting for the overall training costs, which was a key goal of our study. The inefficiency of lottery tickets has been shown in [1], which finds that rewinding the weights performs worse than learning rate restarts. Additionally, [2, 3] provide explanations on why lottery tickets fail at scale.
>
> * [1] https://arxiv.org/abs/2003.02389
> * [2] https://arxiv.org/abs/2010.03533
> * [3] https://arxiv.org/abs/2210.03044
>
> > 2. Can the proposed scaling laws be useful in the extreme sparsity regime, e.g. 95% or even 99%? Or is it applicable only for the sparsity level within a range?
>
> We did initially also perform a few experiments for 93.75% sparsity (16x compression); however, as also suggested by, e.g. Figure 2, improvements were diminishing and producing those models quite expensive (we have to train 16x larger models to eventually reach a certain target nonzero count). Thus, we decided to focus our computational budget on the most interesting and practical range. In principle, our scaling laws should also extend to higher sparsities (though, one should be careful with extreme extrapolation for any kind of scaling law due to potential problems like training instabilities for very unusual configurations).

---

> ### Comment · Reviewer_1amU · 2023-11-21
>
> The rebuttal have addressed my concerns. I appreciate the authors for adding the experiments on decoder-only Transformers. Thus I maintain my positive assessment of this paper (confidence is increased to 5).
>
> I noticed that the term "computational budget" is mentioned a lot. I would suggest give some details on this, e.g. hardware and training time. In this way, the reader may have a concrete idea on the large-scale nature of the conducted experiments.

---

> > ### Author Response · Authors · 2023-11-21
> > **Author Response**
> >
> > We thank the reviewer for their response! We will include more information on computational costs in the final version of the paper. We are glad that our rebuttal has addressed the reviewer's concerns and that the reviewer confidently maintains their positive assessment of our work. We see the score is kept at "6: marginally above the acceptance threshold". Is this intentional? If so, we would appreciate additional feedback for improving our work further. We thank the reviewer again for their time.

---

> > > ### Comment · Reviewer_1amU · 2023-11-21
> > >
> > > I think the results on decoder-only Transformers should be highlighted in the main paper, instead of the T5 encoder-decoder architecture. The original scaling law study [1] was conducted on such autoregressive models. It would also fit more with the use of "foundation models" in the paper title. This is my main reservation for not increasing my score.
> > >
> > > [1] Scaling laws for neural language models. 2020

---

> > > > ### Author Response · Authors · 2023-11-22
> > > > **Author Response**
> > > >
> > > > We mainly put the decoder-only results into the Appendix to make them easy to view during the rebuttal period. We agree that, due to their popularity, having decoder-only model results in the main text would further improve our work. To this end, we will integrate our most recent results into the main text and have started running additional decoder-only experiments (n:m sparsity) as well. The decoder-only results we shared during the rebuttal are in line with our previous encoder-decoder experiments, showing that our conclusions remain the same. We hope this addresses the remaining concern of the reviewer and we would like to thank them again for their time.

---

### Official Review · Reviewer_jJsH · 2023-11-02

**Soundness:** 3 good
**Presentation:** 4 excellent
**Contribution:** 3 good
**Rating:** 8
**Confidence:** 4

**Summary:**

The key message of this paper is a scaling law describing the relationship between sparsity (weight sparsity rate), the number of nonzero parameters, and the amount of training data proposed. The most intuitive effect on our training of sparse models is that for a fixed number of non-zero parameters, it is found that optimal sparsity increases with the amount of data used for training. And the benefit of long training seems more obvious compared with dense models. The experiments are using Vision Transformer and T5 language models and conducted on large-scale C4 and JFT4B datasets.

**Strengths:**

This study is the first paper to use large-scale experiments and theoretical analysis to explore in detail the effects of sparsity on neural network training and efficiency. The authors propose new scaling laws describing the relationship between sparsity, the number of non-zero parameters, and the amount of training data, which is important.
On the one hand, the findings of this paper can help us configure the training parameters more scientifically and thus improve the efficiency and performance of the model in large-scale training. It also shows that as the amount of training data increases, the model sparsity required to achieve optimal performance also rises. In this case, appropriately increasing the model sparsity and optimizing the model structure may be a more effective solution.
Meanwhile, according to this finding, the sparsity of the model can be dynamically adjusted according to the change in data volume during the training process to balance the model performance and computational cost better.

**Weaknesses:**

Overall, I think this is a good paper; my only concern lies in the scale of the data and modeling. Compared to Chinchilla's law, the author's data and model sizes are significantly smaller. Also, since the authors mentioned the optimal sparsity setting as an empirical design or product of this claim, could you elaborate on what specific improvements in downstream task or operational efficiency the sparsity of our actual model would bring us under the authors' theoretical guarantee?

**Questions:**

Please see the weakness.

---

> ### Author Response · Authors · 2023-11-17
> **Author Response**
>
> Thank you for the review! We respond to your comments and questions in detail below.
>
> > Overall, I think this is a good paper; my only concern lies in the scale of the data and modeling. Compared to Chinchilla's law, the author's data and model sizes are significantly smaller.
>
> We would like to note that the Chinchilla paper is a highly expensive large-scale study, which was conducted only after the great potential of scaling had been very clearly demonstrated by models like GPT3 or Gopher. The seminal scaling work by Kaplan et al. (2020) was based on smaller model sizes, which are close to ours.
>
> We would also like to highlight a few aspects that make our study quite computationally demanding, already at the model sizes we consider (we discuss this further in Appendix A):
>
> * Producing a model with sparsity $S$ and $N$ nonzeros parameters requires actually training and pruning an $N / (1 - S)$ times larger model (e.g., 8x larger for 87.5% sparsity).
> * We have to sweep not only over model size N and amount of training data D, but also sparsity S, multiplying the number of runs required to collect data for fitting scaling laws by the number of sparsity levels.
> * As shown by our work, sparse models become optimal in longer training regimes, hence we want to include several runs in our fitting data which train significantly longer than Chinchilla optimal (for instance, our ViT runs include settings with 1.8B images and correspondingly ~350 billion tokens cost).
> * We study both vision and language models, further increasing the number of experiments.
>
> Nevertheless, we agree that further investigations at even larger scales would be valuable and hope to look into this for future work. However, we believe that the current set of experiments already validate our proposed scaling laws.
>
> > Also, since the authors mentioned the optimal sparsity setting as an empirical design or product of this claim, could you elaborate on what specific improvements in downstream task or operational efficiency the sparsity of our actual model would bring us under the authors' theoretical guarantee?
>
> The main goal of our work is to understand the scaling behavior of sparsity in the context of foundation models, in general, irrespective of hardware or runtime specific artifacts. This should help ongoing and future efforts on practical sparsity acceleration by answering questions such as: (1) which sparsity levels to expect while recovering accuracy, (2) how long sparse models must be trained to recover accuracy, or (3) how large improvements over dense models can be expected.
>
> In the newest revision, we have added Appendix F, discussing related work on practical sparsity acceleration including: acceleration algorithms for CPU inference (Elsen et al., 2020; Kurtz et al., 2020), n:m sparsity (Pool & Yu, 2021) supported on current NVIDIA GPUs (which we explicitly study in Section 3 as well), and custom hardware specifically designed for sparsity (Emani et al., 2023).

---

> > ### Comment · Reviewer_jJsH · 2023-11-21
> >
> > Thanks for your response. My concerns have been solved. I have increased my score.

---

### Official Review · Reviewer_pZ1g · 2023-11-03

**Soundness:** 3 good
**Presentation:** 3 good
**Contribution:** 3 good
**Rating:** 8
**Confidence:** 3

**Summary:**

This paper proposes a formulation of scaling laws for sparsely-connected foundation models. Beginning with the scaling laws proposed by previous works, which are related to the parameter count N and the amount of training data D, the authors add parameter sparsity S into consideration and construct a formulation based on observations and actual data points. After obtaining the formulation, the authors investigate the optimal sparsity and limit performance for training sparse models. Furthermore, the authors consider two practical applications, including N:M sparsity and pruning, to validate the scaling laws.

**Strengths:**

1. This paper investigates a topic highly relevant to current trends in large-scale models, namely the scaling laws of sparse large models. The authors present the formulation of scaling laws along with their derivation process and validate them on T5 and ViT models.
2. The authors also validate several applications related to scaling laws, providing reasonable experimental designs and detailed experimental results.
3. The writing throughout the article is quite commendable.

**Weaknesses:**

1. This paper posits that sparsity can reduce training costs, thus presenting research on optimal sparsity. However, in reality, the training costs of current sparse training methods are often the same as those without sparsity. Under these circumstances, is it meaningful to study optimal sparsity?

**Questions:**

Please refer to the weakneses.

---

> ### Author Response · Authors · 2023-11-17
> **Author Response**
>
> Thank you for the review! We respond to your comments and questions in detail below.
>
> > This paper posits that sparsity can reduce training costs, thus presenting research on optimal sparsity. However, in reality, the training costs of current sparse training methods are often the same as those without sparsity. Under these circumstances, is it meaningful to study optimal sparsity?
>
> There are already some promising results for achieving practical training speedups through unstructured sparsity, both via hardware accelerators (Emani et al. (2023), Figure 7) as well as sparsity-aware backpropagation algorithms on commodity hardware (Nikdan et al. (2023)). Nevertheless, we agree that this is still an ongoing area of research, which is exactly why we considered two different cost models in our optimal sparsity analysis:
>
> * Dense: no acceleration from sparsity during training whatsoever.
> * Sparse: sparsity is fully exploited as soon as it appears.
>
> In both cases, sparse models become optimal (in terms of size / inference cost) as the training budget is increased; however, this happens more quickly if sparsity can already be exploited during training. These results can also be adapted to different cost models.
>
> In the most recent revision of our paper, we have added additional discussion about related works for utilizing sparsity in practice, for inference speedups, training acceleration and memory reduction (see Appendix F). We believe that our work will be particularly valuable for future work in this area, by providing guidance on optimal sparsity levels, possible accuracy gains and scaling behavior with respect to model sizes and training data.
>
> * Nikdan et al., SparseProp: Efficient Sparse Backpropagation for Faster Training of Neural Networks, 2023.
> * Emani et al., A Comprehensive Performance Study of Large Language Models on Novel AI Accelerators, 2023.

---

> > ### Comment · Reviewer_pZ1g · 2023-11-22
> >
> > Thanks for your response. I will maintain my score in the first review round.

---

### Author Response · Authors · 2023-11-21
**Discussion Period Ending**

Dear Reviewers,

thank you again for your useful comments and suggestions!

We wanted to send a gentle reminder that the discussion period is ending shortly. During the rebuttal, in addition to responding to the reviewers' questions, we have run further experiments using popular decoder-only models trained with auto-regressive language modeling loss (see Appendix F), following the suggestions from multiple reviewers. With this, we now demonstrate the robustness of our proposed sparse scaling laws in 3 different settings.

Since the deadline is approaching, we would be happy to hear back from you and would appreciate it if you considered updating your scores in the light of our discussions.

With best regards,

The authors

---

### Meta-Review · Area_Chair_zr1Z · 2023-12-08

**Metareview:**

This paper delves into a significant and timely area of research, examining the effects of parameter sparsity on the scaling behavior of Transformers in both vision and language domains. The paper is notable for several strengths, as acknowledged by the reviewers:

+ Novelty in Theoretical Exploration: The study is pioneering in its theoretical analysis of the relationship between weight sparsity, non-zero parameters, and training data volume. This is a substantial contribution to the field, offering a new perspective on scaling laws in the context of sparse large models.

+ Large-Scale Empirical Validation: The paper presents extensive empirical results, validating the proposed scaling laws on major models like T5 and ViT. This empirical approach lends significant credibility to the theoretical claims and provides practical insights.

+ Comprehensive Experimental Design: The experiments are well-designed and thorough, covering various aspects related to scaling laws. This thoroughness in experimental design underpins the paper's robust findings and conclusions.

+ Quality of Writing: The paper is commended for its clarity and coherence in writing, which enhances its accessibility and understandability.

+ Practical Implications and Insights: The findings on optimal sparsity and its dynamic adjustment relative to training data volume are particularly insightful. These insights could guide more efficient and effective configurations for large-scale model training.

+ Investigation into Hardware-Friendly Sparsity: The inclusion of experiments on structured sparsity, particularly hardware-friendly configurations, adds a practical dimension to the research.

+ Valuable Theoretical Contribution: The derivation of optimal sparsity for a given inference size and training budget is a key theoretical advancement, providing a useful framework for understanding and implementing sparsification.

**Justification For Why Not Higher Score:**

- Data and Model Scale Concerns: One of the reviewers raised concerns about the scale of data and models used in the study, particularly in comparison to existing works like Chinchilla's law. While the paper's contributions are significant, the relatively smaller scale of data and models might limit the depth and breadth of the insights, especially in the context of larger and more complex datasets and models prevalent in the field.

**Justification For Why Not Lower Score:**

Shown in the meta review.

---

### Decision · Program_Chairs · 2024-01-16

Accept (spotlight)